# Exfoliation of Molecular Solids by the Synergy of Ultrasound and Use of Surfactants: A Novel Method Applied to Boric Acid

**DOI:** 10.3390/molecules29143324

**Published:** 2024-07-15

**Authors:** Sara Calistri, Alberto Ubaldini, Chiara Telloli, Francesco Gennerini, Giuseppe Marghella, Alessandro Gessi, Stefania Bruni, Antonietta Rizzo

**Affiliations:** 1ENEA, Italian National Agency for New Technologies, Energy and Sustainable Economic Development, C.R. Bologna, Via Martiri di Monte Sole 4, 40129 Bologna, Italy; sara.calistri2@unibo.it (S.C.); chiara.telloli@enea.it (C.T.); giuseppe.marghella@enea.it (G.M.); alessandro.gessi@enea.it (A.G.); stefania.bruni@enea.it (S.B.); antonietta.rizzo@enea.it (A.R.); 2Department of Pharmacy and Biotechnology, University of Bologna, 40126 Bologna, Italy; 3Department of Electrical, Electronic and Information Engineering “Guglielmo Marconi” (DEI), Biomedical Engineering, Cesena Campus, University of Bologna, Via dell’Università 50, 47522 Cesena, Italy; francesco.gennerini@studio.unibo.it

**Keywords:** boric acid, liquid exfoliation, surfactants, nanoparticles

## Abstract

Boric acid, H_3_BO_3_, is a molecular solid made up of layers held together by weak van der Waals forces. It can be considered a pseudo “2D” material, like graphite, compared to graphene. The key distinction is that within each individual layer, the molecular units are connected not only by strong covalent bonds but also by hydrogen bonds. Therefore, classic liquid exfoliation is not suitable for this material, and a specific method needs to be developed. Preliminary results of exfoliation of boric acid particles by combination of ultrasound and the use of surfactants are presented. Ultrasound provides the system with the energy needed for the process, and the surfactant can act to keep the crystalline flakes apart. A system consisting of a saturated solution and large excess solid residue of boric acid was treated in this way for a few hours at 40 °C in the presence of various sodium stearate, proving to be very promising, and an incipient exfoliation was achieved.

## 1. Introduction

One of the greatest innovations in the field of materials science in recent decades was the discovery of so-called two-dimensional (2D) materials, i.e., those materials that are made up of a single layer of atoms [1]. Their thickness is therefore negligible compared to the size they can have in the other two dimensions, which can be tens or hundreds of microns in most cases, but also much higher for some specific materials.

The first of the 2D materials was graphene [2], which is also the most famous of this family of compounds. It is an allotrope of carbon consisting of a single layer of atoms arranged in a hexagonal lattice nanostructure. From a certain perspective, graphite can be seen as a three-dimensional combination of graphene sheets held together by van der Waals (vdW) forces [3]. Graphene has extraordinary chemical–physical properties [4]. Graphene’s conduction and valence bands have a linear dispersion and cross at the Fermi level in the first Brillouin zone, which are Dirac points [5]. Hence, graphene is a zero-gap semiconductor. Holes and electrons near the Dirac point behave as massless fermions (m* = 0) and travel at extremely high speeds (10^6^ m/s) [6]. Graphene displays remarkable electron mobility at room temperature, with reported values in excess of 15,000 cm^2^·V^−1^·s^−1^. With regards to the graphene resistivity, at room temperature it is 10^−8^ Ω·m, lower than silver and copper resistivity at the same temperature. Furthermore, graphene has exceptional thermal conductivity and is nearly transparent (97.7% of light). It has a theoretical tensile strength of 130 Gpa, which is much higher than any steel [7].

Graphene is still probably the most important 2D material but no longer the only one. On the contrary, it is regarded as the progenitor of a new class of materials. Today, numerous other substances share the distinctive characteristic of being two-dimensional (2D). Among them, other allotropes of carbon should be cited: graphyne and graphene [8]. Other elements such as: B, Si, Ge, P, Sn… can exist in 2D form: borophene [9], silicene [10], germanene [11], phosphorene [12], stanene [13], respectively.

Some 2D compounds exist such as hexagonal boron nitride nanosheets [14] and, in a broader sense, many two-dimensional metal chalcogenides [15] such as MoS_2_ [16] or WSe_2_ [17], or topological insulators like (Bi,Sb)_2_(Se,Te)_3_ [18] and many others, including oxides [19] and hydroxide [20]. For these compounds, instead of a single atomic plane, more or less flat or wrinkled, the single 2D unit corresponds to a single crystallographic layer or block of the three-dimensional (3D) bulk material.

The defining characteristic shared by all these materials is that their bulk crystals are layered, made up of planes or blocks where the bonds between the atoms are strongly held together by weaker van der Waals forces, and they are collectively defined as vdW crystals. From a certain perspective, the individual crystalline layers or blocks can be seen as infinite-sized molecules.

Most of these ultrathin materials have very promising properties, often due to peculiar quantum effects, and sometimes their discovery has represented a revolution in solid-state science. They have enhanced physical, chemical, and biological functionality and find interesting technological applications in many fields, from modern microelectronics to photovoltaic systems, to the biomedical field, for issue engineering, contrast agents, bioimaging, and others [21].

As regards their synthesis, there are two types of approaches: bottom-up, i.e., materials grow starting from smaller units, atoms, molecules, or clusters; or top-down, i.e., from the exfoliation of their three-dimensional equivalents [22]. The former methods generally lead to the formation of samples of excellent quality but have the important limitation that they are often expensive, slow, and can be used for small-sized samples. The second method, on the other hand, is potentially more industrializable, but sometimes the quality is lower and less controllable.

Mechanical exfoliation techniques, in particular the so-called “scotch tape” method [23], are often used, but although these methods are easy, very practical, and inexpensive, they are not completely repeatable. Alternatively, chemical exfoliation, using solvents, is, in many cases, the best compromise. Among these methods, liquid-phase exfoliation (LPE) is a liquid-phase method that is used to break a layered crystal into 2D materials by suspending the crystal in a “solvent” and blasting the crystal with ultrasounds [24]. When a powdered layered crystal is dispersed in an appropriate solvent, it can be exfoliated in small pieces if the operating conditions are correct. In this way, graphite and other materials can be converted into large quantities of nanosheets. In general, these nanosheets tend to be a few monolayers thick and of lateral sizes reaching many microns. The solvent must possess the capability to adequately weaken interlayer forces, yet it should not be so effective as to fully dissolve the material and form a solution.

The number of van der Waals (vdW) crystals from which 2D materials can be derived is potentially greater than those currently identified. This underscores the significant research importance in exploring these materials. However, it is uncertain whether effective exfoliation methods exist for all vdW crystals, particularly when the intralayer bonds are not notably strong. This limitation arises because specific liquids, often organic solvents with considerable toxicity, or combinations thereof, are frequently required, presenting challenges in the development of universal exfoliation techniques for diverse vdW crystals [25]. It is not certain that these solvents are available for each layered compound, and consequently, some materials are excluded from the possibility of forming 2D systems by this method.

This is the case of boric acid (H_3_BO_3_). It can be seen as the simplest of the borates, and it is an interesting material with many applications. It is an antiseptic and an antifungal [26], it is used as a fire retarding agent [27] and as a lubricant [28]; it has applications in the field of the nuclear industry because it can effectively regulate the fission speed that occurs in pressurized water reactors (PWRs) [29]. Due to its chemical properties, it is not possible to prepare nanoparticles following traditional methods used for metals like gold, silver, or copper, such as controlled reduction in aqueous solution. In this case, the nano/microparticles are prepared by fine grinding [30] or by freeze-drying [31]. However, using these methods often results in particles with a wide size distribution and frequently of low crystallographic quality.

Although at room pressure hexagonal [32] and trigonal [33] polymorphs exist, the most commonly appearing H_3_BO_3_ crystallizes in triclinic system, space group P-1, cell parameters: a = 7.0187 Å, b = 7.0350 Å, c = 6.3472 Å, α = 92.49°, β = 101.46°, γ = 119.76° [34], with a structure consisting of layers of planar B(OH)_3_ molecules held together by hydrogen bonds. This presents an intriguing case, as the intralayer forces are not substantially stronger than the interlayer forces.

Boric acid is soluble in water, ethanol, glycerol, and slightly soluble in acetone, but not in most non-polar organic solvents, such as ethers. Classic LPE would involve identifying a solvent capable of weakening interlayer bonds, but not intramolecular ones, which is difficult because the energy difference is not as marked as between covalent bonds and vdW bonds. In this case, the theoretical ideal liquid should reduce vdW forces but not disturb hydrogen bonds. Such a liquid or mixture of liquids could not exist, and in any case, a laborious and long quest would be necessary to identify it. LPE also requires, in most cases, the use of non-concentrated dispersions of the material to be exfoliated in the suitable liquid.

Therefore, classical LPE may not be suitable in the case of boric acid and other possible analogous materials. It must be at least modified ad hoc to obtain nanosheets or at least to obtain nanometric-sized particles.

It was therefore decided to follow a different approach, namely, to use a natural solvent of the compound, i.e., water or ethyl alcohol, but in small quantities in order to prepare a highly supersaturated system. In this way, most of the compound remains in the solid state, but one can imagine that there is a dynamic equilibrium, thanks to the sonification action, that causes the particles to flake off. The addition of a surfactant can allow the layers to be separated, and, at the same time, it can permit the formation of a stable dispersion. In this case, the effects of sodium stearate (C_18_H_35_O_2_Na) and sodium lauryl sulfate (C_12_H_25_SO_4_Na) were tested. At the end, the surfactant can be removed chemically by washing with organic solvents, for example, hexane, obtaining the separated nanosheets.

Investigations were carried out using characterization techniques, such as Raman spectroscopy with an optical Raman microscope, X-ray diffraction, and scanning electron microscopy (SEM-EDX).

## 2. Results and Discussion

Figure 1A shows the crystals’ structure of the boric acid, oriented to highlight its layered organization, and Figure 1B shows the top view of a single plane, for underscoring the covalent (solid lines) and the H-bonds (dotted lines).

Each boron atom is surrounded by three oxygen atoms forming triangular BO_3_ groups. Because oxygen has a much higher electronegativity than boron, these bonds have a partial ionic character. The bond length B–O is about 0.13 nm, and the distance among the oxygen atoms is about 0.23 nm [34,35]. The boron atoms are arranged in zig–zag chains. Oxygen atoms instead are placed at the corners of regular hexagons where the hydrogen atoms lie asymmetrically along the sides, because each of these atoms is covalently bound to an oxygen atom and by an H-bond to another. Each of these hexagons is separated from the others by two boron atoms, so that layers result to be formed by isolated hexagonal rings, each of which is surrounded by six smaller, elongated hexagonal rings. The interlayer distance is about 0.32 nm.

Typically, the energy of a hydrogen bond is of a few ten kJ/mol (about 21 kJ/mol for the water), i.e., about 5% the energy of covalent bond O–H, that is, 464 kJ/mol [36]. van der Waals energies are normally in the range of a few tenths to few kJ/mol [37], so still from twice to more than ten times weaker than H-bonds; therefore, in principle, intralayer forces should be strong enough to allow exfoliation of the material.

Despite this, the solubilization of boric acid in water or ethanol leads to the complete dissolution of the crystals and the formation of free, isolated molecules in solution. Boric acid is a very weak acid with a pK_A_ value of 9.2 [38]. At a lower pH than 7, it is present in its non-dissociated form.

It does not seem realistically reasonable to obtain nanosheets from dilute boric acid solutions because the molecules are randomly dispersed in the solvent (called in the following S1), and recrystallization eventually leads to the formation of three-dimensional crystals, of relatively large size. Figure 2 shows the SEM images of some crystals obtained by simply slowly cooling a solution from a high temperature to room condition. The solubility of boric acid increases strongly as the temperature increases [39]. Therefore, by cooling, saturation conditions are soon reached, and the crystals can grow. A scotch tape-type exfoliation technique was attempted on some of these crystals, but it was not possible to iterate the procedure in a sufficient number of times to reach nanometric thicknesses. On the contrary, the crystals pulverize after very few times.

The concept underlying this work is that ultrasound can fragment particles into smaller pieces, and a surfactant can prevent these individual flakes from re-aggregating, thereby hindering crystallization. When these three components are combined, the growth of crystals is effectively inhibited. Sonication alone also does not result in exfoliation of the material, starting from the supersaturated solutions studied in this work. Likewise, surfactant alone does not produce flakes. The selection of the appropriate surfactant is crucial, and it can be expected that there may be some surfactants that exhibit superior exfoliating capabilities compared to others [40,41]. However, scientific references specific to this context are limited, as the LPE method is typically applied to different material categories. In some cases, highly specific molecules are required, while in others, the dispersions of the thin layers are sufficiently stable regardless of the chemical nature of the surfactant. Sethurajaperumal et al. demonstrated the effective exfoliation of graphite using a plant-derived surfactant [42]. Amino acids have been successfully employed in the case of boron nitride [43].

Therefore, the selection of the appropriate surfactant is a critical consideration, and unfortunately, a trial-and-error approach is often necessary in this case. This is due to the limited understanding of the interactions between boric acid and surfactants, which remains a relatively understudied topic.

Surfactants are molecules capable of reducing the surface tension of many liquids and are made up of a hydrophilic polar “head” and a hydrophobic “tail”. Normally, they are classified based on the polar head and are divided into anionic, cationic, or non-ionic. Among these, anionic surfactants seem the most suitable for this intended purpose. Indeed, despite its weak acidity, boric acid could potentially react with cationic surfactants in an acid-base manner, whereas ideally, the role of the surfactant should solely be to facilitate particle dispersion.

Interactions with non-ionic surfactants may be more difficult to predict; however, many of them belong to the class of alcohols or ethers. Among them, there are in fact molecules such as pentaethylene glycol monododecyl ether (C12E5) or octaethylene glycol monododecyl ether (C12E8). Interestingly, boric acid can form coordination complexes, although not particularly stable, with the hydroxyl groups of alcohols [44,45,46]. Therefore, again, these systems may not be suitable for achieving exfoliation.

For these reasons, anionic surfactants appear to be the most natural choice. However, the number of potential molecules remains exceedingly high in principle. Additionally, anionic surfactants can feature diverse functional groups such as sulfate, sulfonate, phosphate, and carboxylates. In selecting which molecules to use, the criterion was to choose commonly employed molecules with low toxicity but varying characteristics, aiming to explore the possibility of exfoliation while gaining a broader understanding. Cost-effectiveness was also a consideration, as scalability and affordability would be crucial for any potential large-scale applications. Sodium stearate and sodium lauryl sulfate respond well to these needs, because they have different functional groups (carboxylate and sulfate anions), different chain lengths, and different chemical–physical properties, such as solubility in water or strength of the conjugated acid.

This preliminary work aims to demonstrate the possibility of achieving the goal of exfoliation. Many outstanding questions remain, for example, the effect of chain length, and other chemical properties of surfactants, and further research will be necessary to clarify these aspects.

Even so, however, starting from a dilute solution, it is reasonable to expect that boric acid crystals will eventually form. Therefore, the starting point should be an extremely concentrated solution, or better said, a two-phase system solid boric acid-saturated solution of boric acid in a solvent. At the end of the process, the problem remains of separating the boric acid particles from the surfactant. This can be solved by adding a solvent (S2) in which the acid and water or ethanol are completely insoluble and in which the surfactant is soluble. In this way, it is possible to obtain a phase separation between two liquids of which one is rich in H_3_BO_3_ and poor in surfactant and the other is, on the contrary, rich in surfactant but poor in the compound of interest. The two liquids can be separated easily, for example, using a separating funnel.

The method can be in principle effective but poses some problems that need to be resolved. The first is the choice of S1, i.e., water or ethanol, capable of solubilizing both the boric acid and the surfactant, to make the exfoliation process take place, and S2, necessary for the separation. It must be a non-polar liquid, such as ethers like diethyl ether, or liquid alkanes, such as hexane. This constraint limits the possibilities. In fact, some pairs of solvents, such as ethanol and hexane, have too high a mutual solubility, and furthermore, the presence of the surfactant could induce the formation of emulsions. Therefore, it may be impossible to separate the two liquids. For this reason, it was decided to use water as S1 and ethyl or isopropyl ether as S2. Sodium stearate and sodium lauryl sulfate are among the most used surfactants in chemical processes, but being ionic compounds, they are not soluble in non-polar solvents. However, their respective fatty acids are. For this reason, the pH of the system was lowered after the ultrasonication step by adding a small amount of nitric acid, which is not expected to have any effect on boric acid other than to increase the non-dissociated fraction, but it is able to transform surfactants into their conjugate acid.

Several parameters play a crucial role in this process, including temperature, processing duration, ultrasound parameters such as energy and possibly frequencies, the ratio of boric acid to surfactant, and the excess amount of boric acid beyond its saturation concentration. These factors collectively influence the effectiveness of particle fragmentation and the prevention of crystallization in the system. The effect of each of them should be explored systematically to optimize the process. It should also be kept in mind the possibility that, by changing multiple factors at the same time, the results could be quite different.

In the described method, boric acid must be present in quantities exceeding its saturation concentration at a fixed temperature, as otherwise, it would dissolve completely in water. However, determining the exact degree of supersaturation required is challenging. Preliminary experiments were conducted to gather information, revealing that systems with approximately double the saturation concentration of boric acid result in simple recrystallization and crystal formation. On the contrary, in systems in which the quantity of boric acid was much higher, up to 10 times greater, what was obtained was a rather compact mass, which was not possible to treat with subsequent separation processes. The ideal quantity is around 4–5 times over the saturation limit.

Something similar is observed regarding the amount of surfactant. If it is too low, simply no exfoliation takes place, and it cannot act to keep the sheets separated; if it is too high, the surfactants do not dissolve completely, and it is difficult to remove them from the system. It was chosen, after some attempts with other compositions, to keep the ratio by moles between them and boric acid constant, equal to 1%, because this value represents a good compromise between these two contrasting aspects.

Process temperature and duration are certainly important parameters, but in this work, we chose to keep them constant. Considering that the water solubility of surfactants and boric acid increases with temperature, and in the case of the latter by a lot, all the experiments were conducted at 40 °C, because this temperature is sufficiently low to prevent complete solubilization but high enough to allow sufficiently high kinetics.

In the experimental section, a table summarizes these preliminarily studied parameters.

Despite this, it should be considered that the optimization of the process may require different combinations of several different factors.

Figure 3A–D show, at different magnifications, the initial particles of aH_3_BO_3_ before any treatment (Figure 3A), the particles after sonification without any surfactant present (Figure 3B), and the particles after sonification in the presence of sodium lauryl sulfate and separation by ether (Figure 3C,D).

Initially, particles have a rounded, potato-like appearance; after being sonicated instead, particles appear with a flat and crystalline appearance, with angles of 120°. It could be imagined that this is a recrystallization process induced by the temperature and the energy by the ultrasound. The edges of the crystals, however, exhibit signs of damage and partial breakage, likely resulting from mechanical agitation induced by ultrasound. This agitation causes the particles to collide with each other, leading to these collateral effects on crystal structure.

No isolated flakes or nanoparticles are formed, but nevertheless, Figure 3C,D clearly show the layered nature of this material, as it is possible to observe the presence of very thin, although not separated, crystalline sheets. They suggest that an incipient exfoliation process takes place. The thickness of the layers is indeed much smaller than their lateral size. However, clumps of small particles are evident above the H_3_BO_3_ crystals. The compositional analysis highlighted that they are composed of C, O, S, and Na, i.e., that they are solid particles of the surfactant. However, this could indicate that the method as such has the potential to exfoliate this material and that nanoparticles or single sheets might be prepared.

These observations offer a clue to the mechanism of the exfoliation process at the microscopic level. The ultrasounds provide energy to the system and act as a sort of very strong mechanical agitation [47]. They are evidently capable of demolishing the initial particles but fail to separate the layers from each other, because this requires not only physical action, but also chemical process must take place. Surfactant molecules can perturb the van der Waals forces acting between the layers, probably by being absorbed and intercalating between them and increasing the lattice distance [41,47,48]. However, sodium lauryl sulfate is very soluble and probably not very effective in this action. Therefore, it is only possible to have an incipient exfoliation, not a complete one.

Finally, the subsequent chemical extraction was not complete, likely because in these conditions, it is not possible to completely protonate the anion in solution, causing a certain quantity to remain in the aqueous phase which then forms the small sulfur-rich particles. Therefore, this specific surfactant does not seem to be suitable for this purpose, but this indicates that others could act better, and in fact, this is what is observed in the case of sodium stearate.

Using this surfactant, a clear separation interface between the two liquids was formed in the separating funnel, and the solid residue accumulated completely in the lower part. In this case, a part of the solid particles was floating in the aqueous phase, until at the interface with the organic phase. This may indicate that many particles are small. In fact, the morphological analyses shown in Figure 4A,B showed rather different results.

Even in this case, the exfoliation process is not complete, meaning that the operating conditions were not the most effective possible; however, small and isolated crystals can easily be seen whose thickness is certainly less than a micron. In some larger particles, it is possible to recognize individual layers, which, in turn, are made up of smaller blocks, a few hundred nanometers thick. Their surface is very uniform and smooth. In some cases, there are stacks of overlapping sheets, but they do not appear connected. The lateral size of the crystals appears quite uniform. Very small particles, or very large ones, are rare. This suggests that the exfoliation is rather uniform, and that the surfactant could keep the individual sheets that form separated. No particles of different composition or morphology are evident, indicating that the removal of the surfactant was quite effective.

The optimization of the parameters, particularly the intensity of the ultrasound, has yet to be identified, but despite this, it seems possible to imagine that a more advanced degree of exfoliation could be achieved and even a complete dispersion of isolated thin flakes.

While the samples treated with sodium lauryl sulfate were not further characterized, because it was not possible to remove the surfactant, those treated with sodium stearate were subjected to further chemical–physical analysis to try to investigate the effects of this treatment in more depth.

Figure 5 shows Raman spectra of initial H_3_BO_3_ (a) of the sample after the ultrasonication (b) and in the presence of sodium stearate (c) normalized to the most intense band (which is the one at approximately 880 cm^−1^).

Rather surprisingly, there is a certain paucity of vibrational spectroscopic works on pure boric acid, especially in recent years.

Isolated molecules of H_3_BO_3_ have C_3h_ symmetry, but in solid state, it is lowered to C_6h_ [49,50]. IR and Raman bands are interpreted as skeletal vibrations of the BO_3_ groups plus the hydrogen vibrations of hexagonal O–H–O rings. Due to their structure, these bands are mainly due to intralayer vibrations, as the layers are weakly coupled. In each layer, a “unit-cell” can be recognized, which contains two boric acid molecules. For this, since in this bidimensional unit cell there are 14 atoms, 3 N-3, i.e., 39 vibrational frequencies are expected. For standard group theory, the irreducible representation of group C_6h_, the modes are [49]:Γ = 4A_g_ + 2A_u_ + 3B_g_ + 4B_u_ + 2E_1g_ + 4E_1u_ + 5E_2g_ + 2E_2u_

These vibrations are essentially of two classes: either they involve movements of hydrogen atoms, or they are principally a skeletal vibration with the OH groups vibrating approximately as units [50,51].

According to the literature, Raman active modes are reported in Table 1.

The spectra of all the samples are very similar to each other and indicate that the treatments have not altered the purity of the initial material.

The spectra of the initial material and those treated with ultrasound alone are nearly identical, showing almost complete superimposability. Instead, in the case of the sample also treated with sodium stearate, few differences are observable. In this case, under the same spectra acquisition conditions, the intensities of the peaks are rather weaker, decreasing the signal-to-noise ratio, and there is a certain drift in the background of the spectrum. Often, these conditions are associated with a lower degree of crystallinity, and this may suggest that treatment has somehow disrupted the particles. This does not appear to be in perfect agreement with electron microscopic observations, which, on the contrary, show that the particles are good-quality crystals. Quite often, the bands in the Raman spectra in the case of nanostructured materials normally exhibit significant shifts and broadenings, both greater the smaller the size, compared to those observed in bulks. A similar behavior could be expected in the presence of nanosheets, at least for some bands, but in this case, there are no measurable band shifts or broadening. However, the relative intensity of some bands changes slightly and this is particularly noticeable in the low wavenumber part of the spectrum.

In most crystalline solids, this part of the spectrum, i.e., that one below about 150 cm^−1^, corresponds to collective lattice vibrations [52]. In particular, the lattice modes are either translational motions of the center of mass of the molecules or hindered molecular rotation of blocks or structural sub-units.

Often, in layered materials, such as transition metal chalcogenides, boron nitride, and others, some low-frequency Raman bands are a result of the vibrations of entire layers, and either perpendicular (shear) modes or parallel (breathing) [53]. Shear modes can be conceptualized as the movement of atomic or molecular layers sliding antiparallel to each other, whereas the breathing modes involve the layers moving away from and towards each other. These modes cannot be easily interpreted or calculated from the irreducible representation of that crystalline structure. These modes are sensitive to the presence of amorphous components or to structural defects and to the size of the crystallites.

In the sample treated with ultrasonication in the presence of stearate, the peaks in this region are significantly more intense than those of the starting material. Furthermore, the lower their wavenumber, the broader they are, so that at around 110 cm^−1^, an intense and broad band appears which is weak in the other samples. The drift below this band is also much more pronounced in this sample. This could be related to the size of the crystals and the thickness of the sheets, which are smaller and more numerous in the case of the sample treated using stearate.

The diffractograms of the three samples considered are also very similar to each other, in good agreement with the fact that the initial quality and purity are not altered by the process. In the initial sample, however, there are weak peaks which are due to the presence of boric anhydride, B_2_O_3_, and in the one treated with the stearate, others which are due to small residual traces of surfactant, indicating that some further washing with ether is required. Figure 6 shows the pattern of the same after the ultrasonication step.

The XRD patterns of nanostructured materials are normally quite different from those of the same material in bulk form [54]. Often in the first case, a broadening and possibly a shift of the peaks is observed. The diffraction peaks are generated due to the constructive interference of X-ray reflected by crystal planes. Interference from a large number of planes as in bulk crystals is very sharp, but small crystallites, with size lower than few tens of nanometers, possess a restricted number of reflection planes, causing wider diffraction peaks. Indeed, preferential orientations of nanoparticles can induce changes in the relative intensities of all peaks compared to those expected based on the diffraction pattern of the bulk powder. However, it is important to underline that this does not necessarily concern all the peaks present in the diffractogram in the same manner and, in the case of nanosheets which are much smaller in one dimension than in the others, it mainly concerns the peaks that have Miller indices corresponding to the crystallographic direction where the dimensional reduction occurs [55,56].

The insets of Figure 6 show the normalized XRD patterns near the most intense peak and the part at low angle of samples of after ultrasonication without and with sodium stearate, in order to highlight the peaks corresponding to the (0 0 1) and (1 0 0) reflections. While the pattern of the sample that underwent only the ultrasound effect shows peaks with relative intensities very similar to the theoretical ones, in the other case, a variation can be noticed, and the intensity of the peak (0 0 1) is greater. This indicates an incipient preferential orientation. This peak and the most intense one, at about 28°, are also slightly wider. Overall, the signal-to-noise ratio is lower for the second sample because all peaks have lower absolute intensity. The peak (1 0 0) seems to have a significant shoulder, but this is most likely linked to instrumental effects. These results are in agreement with the fact that the sample treated with stearate has a greater number of small and independent crystalline particles compared to the other.

Even though it is evident that the treatment did not produce only independent nanosheets, these analyses indicate that the method has the potential to enable their successful production and that incipient exfoliation occurs.

The different results obtained in the case of lauryl sulfate and sodium stearate, the other conditions being constant, depended on the nature of these molecules.

Lauryl sulfate is more soluble than stearate, also having a shorter chain, and its conjugated acid has a more acidic functional group; therefore, it tends to disperse more in the aqueous phase. This may make it less effective at keeping the thin layers apart.

It may still be difficult to draw definitive conclusions; however, it seems, based on these results, that a longer aliphatic chain, a weaker conjugated acid, and a lower water solubility could be positive for achieving good exfoliation. Future work will be able to investigate these aspects, for example, using surfactants derived from fatty acids with different chain lengths, which also allows solubility to be controlled.

## 3. Experimental Section

Boric acid, 99.8%, sodium stearate, 99%, and sodium lauryl sulfate, 98.5%, from Sigma Aldrich (St. Louis, MO, USA) were used in this work. Some preliminary experiments were carried out to gain information regarding the ratios between materials needed to achieve exfoliation. The temperature was kept constant at 40 °C, and the duration of the process was equal to 5 h. The amount of boric acid was varied between 1.1 times the saturation concentration and 10 times as much. The ratio between the moles of surfactant and those of boric acid was varied between 0.1% and 5%. Table 2 summarizes the conditions used.

Based on these preliminary results, an amount of boric acid in excess of 400 mol% compared to the saturation concentration at room temperature was weighed (the solubility of boric acid at 20 °C is 46.5 g/L [48]) in three different batches. One of them was subjected to ultrasound only, whereas to the other two, 1% by moles of surfactant with respect to boric acid was added, prior to the ultrasonication. The volume of water used in each case was 20 mL. The so-prepared mixtures were put in a sealed plastic containers, in order to prevent losses or contaminations from outside.

Subsequently, these mixtures were ultrasonicated in a commercial ultrasonic bath for 5 h, at 40 °C. (ultrasound power 90 W, frequency 40 kHz). At the end of the process, the system resulted formed by a mass of aggregates precipitated on the bottom of the container, even if few particles were dispersed in the liquid phase which, therefore, was not completely clear.

At this point, to the two mixtures containing surfactants, 1 mL of concentrated nitric acid was added and stirred vigorously. Then, these systems were transferred in a separating funnel and washed at least three times using ethyl ether.

After the separation, the solid residue was carefully dried in an oven at 60 °C.

Figure 7 shows a representation of the process scheme for the preparation of boric acid nanoparticles.

The characterization of the samples, morphology, and composition has been performed by Raman spectroscopy and XRD diffractometry.

Raman spectra of the compounds and mixtures were acquired, at room temperature, by a BWTEK i-Raman plus spectrometer (B&W Tek, Plainsboro, NJ, USA) equipped with a 785 nm laser in the range of 100–3500 cm^−1^ with a spectral resolution of 2 cm^−1^. The measurement parameters, such as acquisition time, number of repetitions, and laser power, were selected for each sample to maximize the signal-to-noise ratio. The standard acquisition was 20 repetitions of 10 s each. This instrument has a maximal power of 350 mW, but in most cases, only 10% of it was used. For each spectrum, a reference acquisition was previously carried out with the same parameters to subtract the instrumental background.

X-ray powder diffraction investigations were performed to determine the crystalline phases, using a Philips X’Pert PRO 3040/60 diffractometer (Philips, Amsterdam, The Netherlands) operating at 40 kV, 40 mA, with Bragg–Brentano geometry, equipped with a Cu Kα source (1.54178 Å), Ni-filtered, and with a curved graphite monochromator. PANalytical High Score software (version 4.1) was used for data elaboration. The XRD acquisitions were performed using these parameters: start position: 10.0125° [2θ]; end position: 99.9875° [2θ]; step size: 0.0250°; scan step time: 6.0000 s, scan type: continuous.

The characterization, morphology, and composition of the samples were performed by scanning electron microscopy (SEM-FEI Inspect-S, FEI Company Hillsboro, OR, USA) coupled with energy-dispersive X-ray spectroscopy (EDX, Oxford Xplore, Oxford Instruments plc, Abingdon, UK). Observations were carried out at different magnifications using both secondary electrons and backscattered electrons detectors at 10 mm working distance, with energy ranging from 10 to 20 KV. The elemental analysis was carried out in the most significant areas of the samples. Data were processed by the software Oxford AZtec One (AZTecLive 6.1 platform).

## 4. Conclusions

Boric acid samples were treated with different types of surfactants under the action of ultrasound. Although complete exfoliation was not achieved, the method appears promising for achieving this result. This is interesting because this material has a layered structure, like graphite and many other compounds from which 2D materials can be obtained, but, unlike them, the molecules in the individual layers are held together by hydrogen bonds and not by strong covalent bonds. A methodology capable of exfoliating even molecular solids could greatly expand the number of 2D materials currently existing. The synergy between ultrasound and surfactants seems promising, but there are many parameters that need to be optimized, starting with the choice of the correct surfactant.

In the present work, two different surfactants were used, which were added to highly supersaturated mixtures of boric acid and water.

At the end of the process, we tried to remove the surfactant with an extraction with an organic solvent in which boric acid is insoluble. The pH of the system was lowered to convert the surfactant into the corresponding fatty acid which is soluble in solvents such as ethers. The surfactants chosen were sodium lauryl sulfate and sodium stearate. The method employed was not effective in fully removing the first component, whereas the second component was almost entirely eliminated. The ultrasound/surfactant synergy was found in both cases to be more effective than the use of ultrasound alone to exfoliate the material, because although not completely free, many thin crystalline sheets were formed. Furthermore, in the case of stearate, many isolated particles with a hexagonal shape and very thin were formed. Their thickness is far below 1 micron, being in some cases even a few tens of nanometers.

Therefore, the methodology seems effective for this purpose. The optimization of the operating parameters, (temperature, boric acid stearate ratio, process duration, ultrasonic energy) still needs to be achieved, and further studies are required to achieve the desired result.

## Figures and Tables

**Figure 1 molecules-29-03324-f001:**
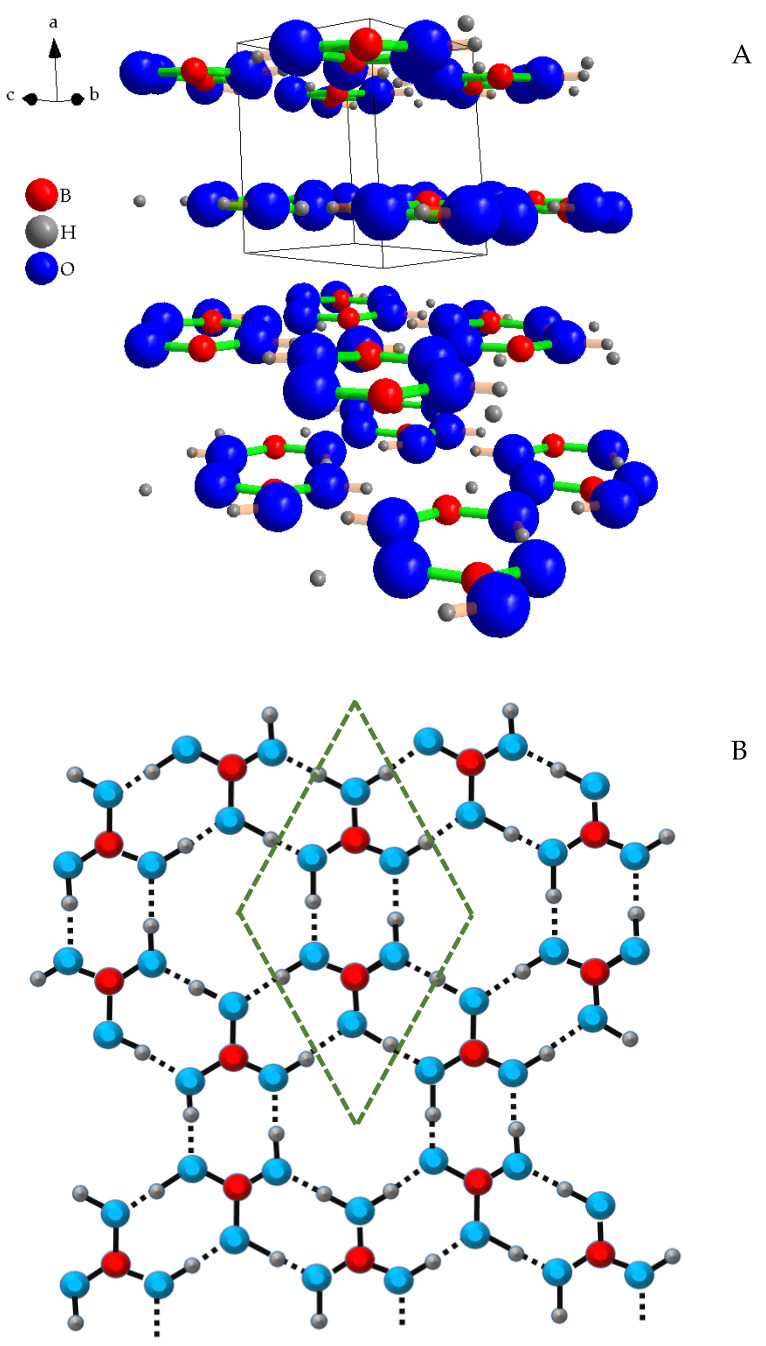
H_3_BO_3_ crystal structure (**A**) (green lines are B–O bonds, light orange lines are H–O bonds), and view of a single layer from above (**B**) (dotted lines are H–O bonds). The dotted green lines outline the “layer unit cell”. “Diamond version 3.2k” software was used to draw the crystal structures.

**Figure 2 molecules-29-03324-f002:**
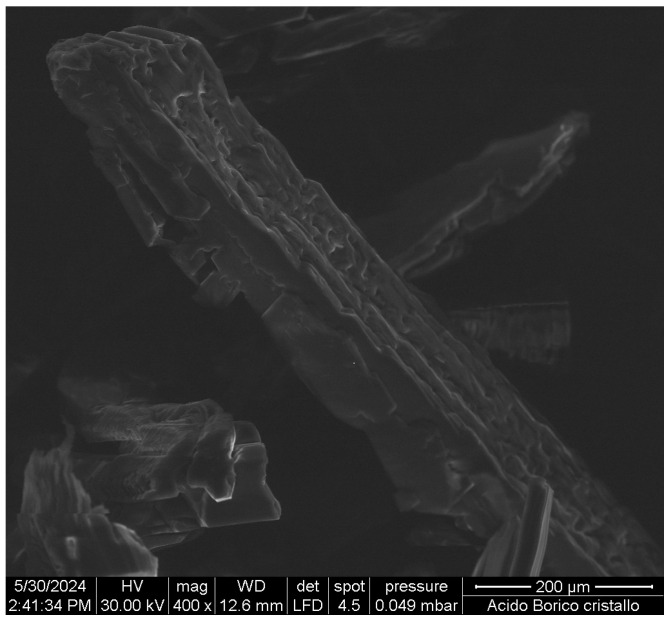
SEM images of crystals of H_3_BO_3_ grown by slow cooling of a hot solution.

**Figure 3 molecules-29-03324-f003:**
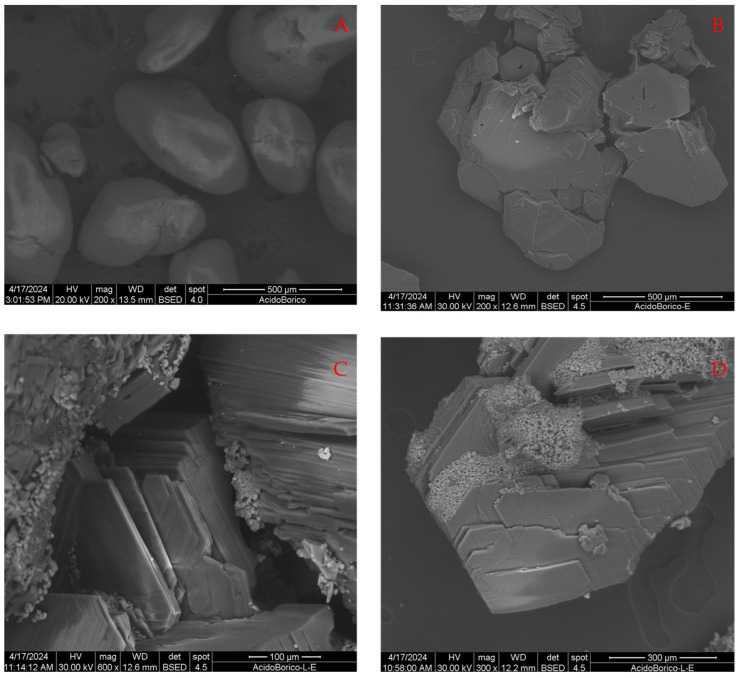
SEM images of the initial boric acid sample (**A**), after undergoing ultrasonication without added surfactants (**B**), and after the process in the presence of sodium lauryl sulfate (**C**,**D**).

**Figure 4 molecules-29-03324-f004:**
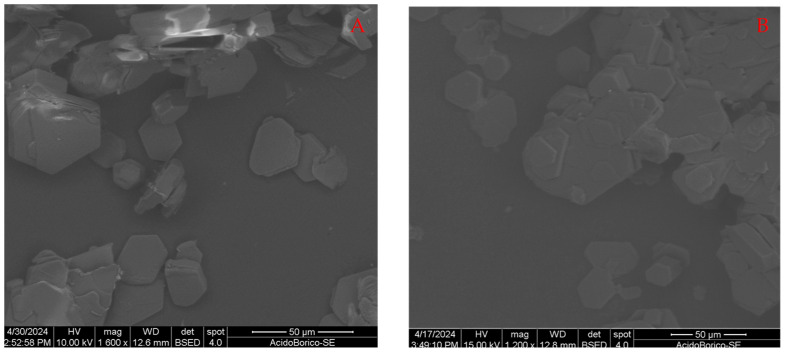
SEM images of crystalline particles of H_3_BO_3_ after the process in the presence of sodium stearate; (**A**,**B**) are different points from the same bacth.

**Figure 5 molecules-29-03324-f005:**
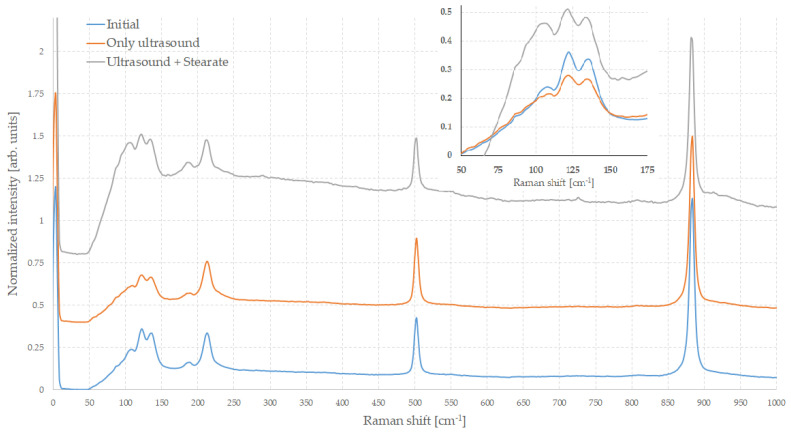
Raman spectra of H_3_BO_3_ samples. Curves are shifted for clarity. The inset highlights the low wavenumbers part of the spectra.

**Figure 6 molecules-29-03324-f006:**
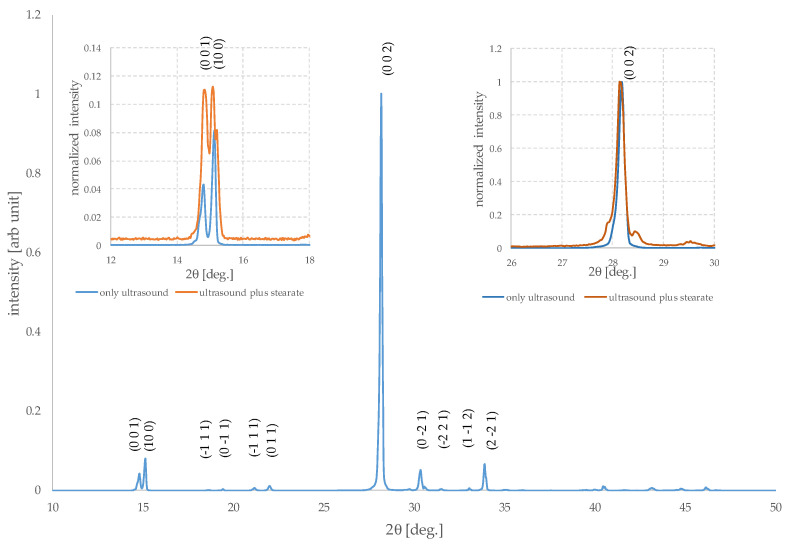
XRD patterns of H_3_BO_3_ sample after ultrasonication. The inset on the right shows the part of normalized XRD pattern near the most intense peak ((0 0 2) reflection) of the same sample, and the one treated with sodium stearate and the inset on the left shows the low angle part of XRD patterns of the same samples.

**Figure 7 molecules-29-03324-f007:**
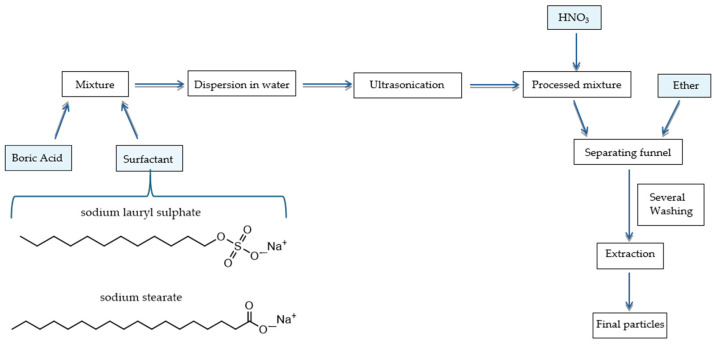
Scheme of operations for exfoliation of boric acid particles.

**Table 1 molecules-29-03324-t001:** Active Raman bands.

Raman Shift [cm^−1^]	Species	Attribution
3251	E_2g_	OH Stretching
3165	A_g_	OH stretching
1384	E_2g_	BO Stretching
1172	E_2g_	BOH Bending
1085	A_g_	BOH Bending
884	A_g_	BO Stretching
735	E_1g_	BOH Bending out of plane
499	E_2g_	OBO Bending
210	E_2g_	Lattice-translatory oscillation
128	E_1g_	Lattice-rotatory oscillation
60	A_g_	Lattice-rotatory oscillation

**Table 2 molecules-29-03324-t002:** Experimental conditions for boric acid exfoliation.

Component	Range	Results
H_3_BO_3_	110% to 200%	Crystallization
	400% to 500%	Initial exfoliation
	600% to 1000%	Formation of a compact mass
Sodium lauryl sulfate	0.1% to 0.5%	No effect
	1%	Incipient exfoliation
	2% to 5%	Residual solid surfactant
Sodium stearate	0.1% to 0.5%	No effect
	1%	Exfoliation
	2% to 5%	Residual solid surfactant

## Data Availability

Data are contained within the article.

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
