# Peer review of "Exfoliation of Molecular Solids by the Synergy of Ultrasound and Use of Surfactants: A Novel Method Applied to Boric Acid"

_molecules, 2024, doi:10.3390/molecules29143324_

Round 1

Reviewer 1 Report

Comments and Suggestions for Authors

The subject of the presented research is very interesting and actual. It is fully consistent with the goals and objectives of the journal. Nevertheless, I cannot recommend the manuscript for publication in its present form. In my opinion, the paper should be reorganized and provide more structured information when presenting and discussing the procedure itself and the results obtained. Below are some of me comments.

1. The manuscript focuses in particular on the effect of surfactants on the exfoliation procedure. Therefore, a brief review of the literature on the use of precursors in the acid exfoliation process would be useful. Why were sodium stearate and sodium lauryl sulfate tested as surfactants?  Is this a new approach by the authors or does such literature data currently exist? The novelty and potential of the work should be clearly and precisely stated.

It seems that the authors synthesized H3BO3 crystals by melting to try the exfoliation technique. If this is the case, it would be useful to provide a brief summary in the Experiments section and the section “Results and discussion”: what were the conditions of synthesis, the temperature of the beginning of cooling, the size of crystals, what were the products of exfoliation and so on.

The authors write that they have previously performed experiments to find the optimal surfactant content. If the results have been published, references are strongly recommended. Otherwise, a brief description of the conditions and compositions would be useful to understand the choice of surfactant concentration.

It would also be interesting and useful to briefly discuss the influence of the main parameters on the process. The authors write that some of them kept constant and some of them changed. Could this information be presented in the form of a table?

Is the figure 3C an enlarged part of the image 3D?

Caption for Figure 4: “SEM images at different magnifications…” – However, images 4 A-C have the same magnification according to the 50 μm scale. Were crystalline particles obtained under different conditions?

Comments on the Quality of English Language

Moderate editing of English language required.

Author Response

Dear Reviewer

The other authors and I would like to thank you for your time and attention to our manuscript and for all your interesting and helpful comments and observations. We think that having tried to answer them has contributed to improving the quality of this work and has made it clearer in many points where there was a certain ambiguity and lack of precision.

In the new version there are all the corrections to the text that were requested. They are well highlighted so as to be easily recognisable.

Here you can read our answers. We have modified our manuscript according to these answers and we have added some new references in order to better support our position.

The manuscript focuses in particular on the effect of surfactants on the exfoliation procedure. Therefore, a brief review of the literature on the use of precursors in the acid exfoliation process would be useful. Why were sodium stearate and sodium lauryl sulfate tested as surfactants?  Is this a new approach by the authors or does such literature data currently exist? The novelty and potential of the work should be clearly and precisely stated.

The idea of trying the exfoliation of boric acid using the method described by us can be considered quite new and there are not many references in the literature in this regard. There are articles in which boric acid is used as an exfoliating agent for other substances, but not about it directly. H3BO3 nano – submicrometric particles exist and they can be prepared by other methods for example strong mechanical grinding or freeze-drying (see for example our reference 31, in the old version), but these methods don’t have precise goal of exfoliate the material.

Furthermore, we had the ambition to show the possibility of exfoliating a new class of materials, namely molecular solids, which are not taken into consideration for this type of process due to their chemical nature. In these solids, in fact, strong intra-molecular covalent bonds are missing. Obviously, each material requires specific conditions, methods, procedures and the topic cannot be covered in a single article, but it is true that if possible to exfoliate one of them (and in a certain sense, boric acid can be seen as the progenitor or at least a reference for them), then in principle it may also be possible to be successful with others.

For this reason, in a certain sense, we had to move a little blindly, because there is no clear starting point.

Having to decide which surfactants to choose, our initial idea was to choose very different molecules and these two surfactants were chosen precisely because they are very different from each other, in terms of aliphatic chain length and functional groups. In common, they have the property of not generating significant quantities of foam at least under the conditions we used.

In our work, exfoliation is a consequence of the joint action of ultrasound and surfactant. Separately, there is no effective exfoliation, if any at all. We expected that there could be important differences between different surfactants and for this reason we used very different molecules, with the aim of "covering" multiple possibilities at the same time. Our idea was to see if it was possible to achieve the goal. Separately, ultrasound or surfactants do not lead to exfoliation, but only their synergic action. However, it was not known what type of surfactant was more effective, whether a long or short aliphatic chain was more useful, what type of functional group worked best.

Sometimes in the literature exfoliation of other materials is obtained using very specific and particular surfactants. For example, this work (Abimannan Sethurajaperumal and Eswaraiah Varrla "High-Quality and Efficient Liquid-Phase Exfoliation of Few-Layered Graphene by Natural Surfactant" ACS Sustainable Chemistry & Engineering 2022 10 (45), 14746-14760) reports the exfoliation of graphite through the use of surfactant derived from a plant. Or in case of hexagonal boron nitride (h-BN), amino acids have been used (Shihao Zheng et al, “Amino Acid-Assisted Sand-Milling Exfoliation of Boron Nitride Nanosheets for High Thermally Conductive Thermoplastic Polyurethane Composites”, Polymers 2022, 14, 4674. https://doi.org/10.3390/polym14214674)

In other cases, very specific surfactants are used, for example Cetyltrimethylammonium bromide, CTAB, or cetyltrimethylammonium chloride (CTAC) or docosyltrimethylammonium chloride (BTAC-228), which however are more complex molecules, more difficult to use and much more expensive.

We tried more commonly used molecules to see if it was possible to exfoliate in some way. We chose different molecules from each other to try to have a more general picture: if we had used CTAB or CTAC we would have obtained less information whether the process had been a success or not. (indeed, they have cationic heads: a priori a reaction with the weak acid H3BO3 cannot completely ruled out)

These surfactants are probably not the best possible solution and more research will be needed to optimize the process. For example, the length of the chain: in the future it will be possible to use surfactants similar to stearate with chains of different lengths.

We tried to edit the text of the manuscript to highlight these concepts more clearly, so that our idea was evident.

It seems that the authors synthesized H3BO3 crystals by melting to try the exfoliation technique. If this is the case, it would be useful to provide a brief summary in the Experiments section and the section “Results and discussion”: what were the conditions of synthesis, the temperature of the beginning of cooling, the size of crystals, what were the products of exfoliation and so on.

This is actually not the case and we apologize if we were not clear enough. Our starting point was commercial boric acid particles. We do not know how they were prepared by the supplier, however their morphology and appearance is that normally observed for this type of product, also from other suppliers. We have not attempted to obtain singles through any treatments other than those described for exfoliation.

The authors write that they have previously performed experiments to find the optimal surfactant content. If the results have been published, references are strongly recommended. Otherwise, a brief description of the conditions and compositions would be useful to understand the choice of surfactant concentration.”

This is our first work on this topic, so we have no references to show. But you are absolutely right: it is useful to show how we arrived at these working conditions. We knew that it was necessary to work beyond the saturation limit of boric acid in water (which also increases greatly with temperature), but we had no idea by how much. Systems just beyond the saturation limit showed no traces of evident exfoliation and on the contrary an incipient crystallization seemed to be necessary, while when we started from systems whose ratio between boric acid and water was still higher than that described, we obtained a sort of very compact slurry. Probably the boric acid to water ratio is a very important factor.

Again we have edited the text for clarity.

It would also be interesting and useful to briefly discuss the influence of the main parameters on the process. The authors write that some of them kept constant and some of them changed. Could this information be presented in the form of a table?

We have added this information to the manuscript, in order to be clearer

Finally, we have tried to answer to your more specific questions

Is the figure 3C an enlarged part of the image 3D?

Caption for Figure 4: “SEM images at different magnifications…” – However, images 4 A-C have the same magnification according to the 50 μm scale. Were crystalline particles obtained under different conditions?”

Actually, figure 3C and 3D show different crystals, even if from the same batch. Concerning figure 4, we were not very precise: we did not mean all the photos at the same magnification. Since the information that can be obtained from these images is clear, to avoid any confusion or misunderstanding we have decided to eliminate some sub-figures and changed the caption

With our best regards

Alberto Ubaldini with the behalf of the authors.

Reviewer 2 Report

Comments and Suggestions for Authors

Reviewer’s Comments

In this study, the authors investigated the effect of combining ultrasonic with different surfactants on the exfoliation of boric acid sheets. The concentration of surfactants were varied to indicate the most appropriate concentration.

The idea is novel and original and the manuscript could be improved from the following comments:

Experimental section:

     1.          Line 555, acid boric should be Boric acid.

     2.          Line 563, the power of ultrasonication should be added.

     3.          Line 569, what is ethylic ether?

Results and discussion:

     1.          There is a difference between sodium stearate and sodium laurylsufate in the functional groups, carboxylate and sulphate anions, respectively. How to explain the difference of efficiency to exfoliate boric acid depending on the functional groups of surfactants.

     2.          To compare between two surfactants, you have to study only one change in the structure which was not applied here. Sodium stearate (C18H35O2Na) and sodium lauryl sulfate (C12H25SO4Na). there is a different in the functional group and alkyl chain length?! can you explain this?

     3.          Particle size using DLS and Zeta potential analysis should be added to compare the stability and interaction of surfactants with boric acid particles.

Author Response

Dear Reviewer

The authors and I would like to thank you for your positive comment on the idea behind our manuscript. We appreciated it very much.

In the new version there are all the corrections to the text that were requested. They are well highlighted so as to be easily recognisable.

We have tried to comprehensively answer the questions you raised and then find the answers in the following lines. Based on these responses, we also modified the manuscript to integrate them into it.

There is a difference between sodium stearate and sodium laurylsufate in the functional groups, carboxylate and sulphate anions, respectively. How to explain the difference of efficiency to exfoliate boric acid depending on the functional groups of surfactants.

To compare between two surfactants, you have to study only one change in the structure which was not applied here. Sodium stearate (C18H35O2Na) and sodium lauryl sulfate (C12H25SO4Na). there is a different in the functional group and alkyl chain length?! can you explain this?”

In our work, two types of surfactants were used and indeed there are notable differences regarding the exfoliation of boric acid particles. It should be considered that there is not a vast scientific literature regarding the exfoliation of molecular solids (in which weak bonds such as hydrogen bonds are predominant) by LPE (liquid phase exfoliation).

Having to decide which surfactants to choose, our initial idea was to choose very different molecules, possibly commonly used and not particularly expensive. This last point is important not so much for laboratory scale tests, but in the best case scenario for any future applications on a larger scale, the cost of raw materials is also decisive.

In the text now we wrote an explanation of the fact that in our opinion the surfactant should be of the anionic type and not cationic or non-ionic. Acid-base reactions between H3BO3, which, although weak, is an acid, and the cationic heads of these surfactants cannot be excluded a priori. Additionally, boric acid can create coordination complexes with hydroxyl groups present in many nonionic surfactants. Therefore the choice of an anionic surfactant seems to be the most natural.

These two surfactants were chosen precisely because they are very different from each other, in terms of aliphatic chain length and functional groups. In common, they have the property of not generating significant quantities of foam. In our work, exfoliation is a consequence of the joint action of ultrasound and surfactant. Separately, there is no effective exfoliation, if any at all. We expected that there could be important differences between different surfactants and for this reason we used very different molecules, with the aim of "covering" multiple possibilities at the same time. However, given that this work could be seen as the first step of a new research, it should be taken into consideration that very different results could be obtained by drastically changing the experimental conditions used. Based on our observations, the use of sodium stearate appears to be a more efficient route than sodium lauryl sulfate. Despite this, honestly, we cannot be sure that, for example, by significantly changing the ratio between the quantities of boric acid and surfactant, the result cannot be reversed. The conjugated acid of sodium stearate is also weaker than that of sodium lauryl sulfate. This can induce notable differences in the concentration of surfactant in solution and we believe it could play a role in the process.

The next steps of our research will be to keep some parameters modified in this work constant and verify their effect. For example, we now believe that sodium stearate is a better choice. The next step could be to keep the functional group constant and change the length of the chain. If indeed the solubility of the surfactant plays a role in the exfoliation process, this could be demonstrated by using some more soluble or less soluble ones and indeed this can be achieved by changing the molecular weight of the molecule, i.e. the length of the aliphatic chain.

In a way, the most important thing was to demonstrate that the process itself is feasible. And we think we succeeded in this point. Most likely - and we are aware of this - there are more effective surfactant molecules. Except that the search to find them will still be extremely long.

Particle size using DLS and Zeta potential analysis should be added to compare the stability and interaction of surfactants with boric acid particles.”

A study of the crystal size distribution curve using laser methods would be extremely useful. And determining the stability of the dispersions would be too.

However, the Z potential depends on numerous factors, temperature, pH, ionic strength and especially the chemical nature of the solvent. All other things being equal, a dispersion might be stable in, for example, water and not be stable in ethanol. This obviously depends on the way in which the electric charges are distributed around the particles.

Unfortunately, in this case, there is a non-negligible difficulty, namely that boric acid is very soluble in water and quite soluble in other polar solvents, such as ethanol. In such solvents, the exfoliated crystals would dissolve.

They would not do this in non-polar solvents, such as organic solvents, but in this case the electrical charges would be less separated. Therefore it may be impossible to determine the Z potential.

With our best regards

Alberto Ubaldini with the behalf of the authors.

Round 2

Reviewer 1 Report

Comments and Suggestions for Authors

The authors have thoroughly commented on all suggestions. Now the manuscript can be published in the journal.

Comments on the Quality of English Language

 Minor editing of English language required.

Reviewer 2 Report

Comments and Suggestions for Authors

Accept